# Murine precursors to type 1 conventional dendritic cells induce tumor cytotoxicity and exhibit activated PD-1/PD-L1 pathway

Megan S. Molina[1,2], Emely A. Hoffman[1], Jessica Stokes[1], Nicole Kummet[1,3], Richard J. Simpson[1,2,4,5], Emmanuel Katsanis[1,2,5,6,7]*

1 Department of Pediatrics, University of Arizona, Tucson, Arizona, United States of America, 2 Department of Immunobiology, University of Arizona, Tucson, Arizona, United States of America, 3 Department of Molecular and Cellular Biology, University of Arizona, Tucson, Arizona, United States of America, 4 Department of Nutritional Sciences, University of Arizona, Tucson, Arizona, United States of America, 5 The University of Arizona Cancer Center, Tucson, Arizona, United States of America, 6 Department of Medicine, University of Arizona, Tucson, Arizona, United States of America, 7 Department of Pathology, University of Arizona, Tucson, Arizona, United States of America

* ekatsani@arizona.edu

**Data Availability Statement:** RNA sequencing data have been deposited to the UA Research Data Repository reDATA and are publicly available at the

## Abstract

The immediate precursor to murine type 1 conventional DCs (cDC1s) has recently been established and named "pre-cDC1s". Mature CD8α+ cDC1s are recognized for suppressing graft-versus-host disease (GvHD) while promoting graft-versus-leukemia (GvL), however pre-cDC1s have not previously been investigated in the context of alloreactivity or anti-tumor responses. Characterization of pre-cDC1s, compared to CD8α+ cDC1s, found that a lower percentage of pre-cDC1s express PD-L1, yet express greater PD-L1 by MFI and a greater percent PIR-B, a GvHD-suppressing molecule. Functional assays were performed *ex vivo* following *in vivo* depletion of CD8α+ DCs to examine whether pre-cDC1s play a redundant role in alloreactivity. Proliferation assays revealed less allogeneic T-cell proliferation in the absence of CD8α+ cDC1s, with slightly greater CD8+ T-cell proliferation. Further, in the absence of CD8α+ cDC1s, stimulated CD8+ T-cells exhibited significantly less PD-1 expression compared to CD4+ T-cells, and alloreactive T-cell death was significantly lower, driven by reduced CD4+ T-cell death. Tumor-killing assays revealed that T-cells primed with CD8α-depleted DCs *ex vivo* induce greater killing of A20 B-cell leukemia cells, particularly when antigen (Ag) is limited. Bulk RNA sequencing revealed distinct transcriptional programs of these DCs, with pre-cDC1s exhibiting activated PD-1/PD-L1 signaling compared to CD8α+ cDC1s. These results indicate distinct T-cell-priming capabilities of murine pre-cDC1s compared to CD8α+ cDC1s *ex vivo*, with potentially clinically relevant implications in suppressing GvHD while promoting GvL responses, highlighting the need for greater investigation of murine pre-cDC1s.

following DOI: https://doi.org/10.25422/azu.data.14241902.

**Funding:** E.K. received funding from PANDA (https://www.azpanda.org/), Courtney's Courage (https://teeupfortots.org/), and Hyundai Hope on Wheels (https://hyundaihopeonwheels.org/). The funders had no role in study design, data collection and analysis, decision to publish, or preparation of the manuscript.

**Competing interests:** The authors have declared that no competing interests exist.

## Introduction

A distinct murine dendritic cell (**DC**) has been identified as the immediate precursor to CD8α+ type 1 conventional DCs (**cDC1s**), termed "pre-cDC1", phenotypically classified as CD24$^{high}$CD8α- [1–5]. Pre-cDC1s become committed to the cDC1 lineage in the bone marrow (**BM**), and migrate out to the periphery where they acquire tissue-specific signals for their maturation into terminally differentiated CD8α+ or CD103+ cDC1s [1, 6–9]. Outside of their differential expression of CD24 and CD8α, pre-cDC1s express the same pattern recognition receptors (**PRRs**) and other receptors as mature cDC1s (TLR3, DNGR1, Xcr1, etc.) [4, 10–14]. CD24 is a membrane glycoprotein that senses damage-associated molecular patterns (**DAMPs**), whereas CD8α does not have any identified function [15, 16]. The expression levels of antigen (**Ag**)-processing enzymes has been compared among DCs and their precursors in the BM and spleen and found that pre-cDCs in the BM express relatively high levels of proteases compare to mature cDCs, but lose this high expression once they migrate to the spleen [17].

Direct comparisons of the two DC subsets have been conducted in the context of viral infection due to the central role of mature CD8α+ cDC1s in cross-presentation and cross-priming of CD8+ T-cells for anti-viral responses. One study found that pre-cDC1s induce greater expansion of Ag-specific memory T-cells and are better at reducing viral load compared to CD8α+ cDC1s [10]. This effect was largely attributed to the longer lifespan of pre-cDC1s in lymphoid organs [10]. It has also been determined that pre-cDC1s are not as efficient at cross-presentation of necrotic, cell-associated material [5, 10]. Further, de Brito *et. al.* reported that pre-cDC1s do not acquire co-stimulatory capacity upon uptake of necrotic cell material, but are still able to cross-present [5]. They also found that, following TLR9 activation with CpG, pre-cDC1s are superior to CD8α+ cDC1s in CD8+ T-cell cross-priming of viral Ag [5].

Aside from anti-viral CD8+ T-cell responses, CD8α+ cDC1s also play a seminal role in promoting anti-tumor CD8+ T-cell responses via cross-presentation and cross-priming. Further, in the context of hematopoietic cell transplantation (**HCT**) CD8α+ cDC1s are recognized as potent suppressors of graft-versus-host disease (**GvHD**) through their ability to induce clonal deletion of alloreactive donor T-cells [18]. To our knowledge, the role of pre-cDC1s in alloreactivity and GvHD, and in generating anti-tumor responses, has not been previously investigated, despite studies documenting their superior ability to cross-prime CD8+ T-cells compared to mature CD8α+ cDC1s [5, 10]. This represents a significant gap in the literature regarding DC function, plasticity, and development in the contexts of transplantation and cancer. Herein we first characterized expression levels of various activating and inhibiting co-stimulatory molecules on pre-cDC1s compared to CD8α+ cDC1s at steady state. We then used a CD8α-depleting antibody to remove CD8α+ cDC1s *in vivo* and then determine pan-DC function in their absence *ex vivo* in the context of alloreactive T-cell stimulation and tumor-killing. Finally, we performed bulk RNA sequencing of cell-sorted pre-cDC1s compared to CD8α+ cDC1s to discern underlying functional differences between these DC subsets.

## Materials and methods

### Ethics statement

Mouse studies were carried out in strict accordance with the recommendations in the Guide for the Care and Use of Laboratory Animals of the National Institutes of Health. Protocols were approved by the Institutional Animal Care and Use Committee at the University of Arizona (IACUC #14–171).

## Mice

All strains of mice used (BALB/c and C57BL/6) were age-matched 6-10-week-old females purchased from The Jackson Laboratory. Mice were housed in specific pathogen-free conditions and cared for according to the guidelines of the University of Arizona's Institutional Animal Care and Use Committee. The method of sacrifice used for these studies was carbon dioxide ($CO_2$) overdose, in accordance with IACUC-approved methods of providing rapid, painless, stress-free death.

## Drug preparation and administration

The CD8α-depleting antibody (2.43 mAb) (BioXCell, BE0061) was diluted in sterile saline for *in vivo* administration. BALB/c mice were injected with 200 μg 2.43 mAb intraperitoneally on day -3 and day -1 relative to the start of the functional assay on day 0.

## Pan-dendritic cell isolation

Spleens were collected from naïve BALB/c mice and single-cell suspensions were generated. Red blood cells were lysed using Pharm Lyse (BD Biosciences). Splenocytes were washed in PBS (HyClone) and then Pan-DCs were isolated using the Pan-DC Isolation Kit (Miltenyi Biotec). Absolute counts were obtained using a hemocytometer and Trypan Blue.

## Flow cytometry

Cells were washed in flow buffer (PBS with 0.5% FBS), incubated with anti-mouse Fc Block (Thermo Fisher Scientific), and flow cytometry was performed as previously reported [19–23]. All antibodies used for flow cytometry are listed in Table 1. T-cell death was determined using Propidium Iodide Ready Flow Reagent (Invitrogen). Fluorescence data were collected using an LSRFortessa cell analyzer (BD Biosciences) and analyzed using FlowJo 2 (Tree Star).

**Table 1. Antibodies used for flow cytometry.**

| Antibody | Clone(s) | Vendor |
|---|---|---|
| Anti-mouse B220 Brilliant Violet 510 | RA3-6B2 | Biolegend |
| Anti-mouse CD4 APC/Cy7 | GK1.5 | Biolegend |
| Anti-mouse CD8α PE-Cyanine7 | 53–6.7 | Thermo Fisher |
| Anti-mouse CD11c FITC | N418 | Miltenyi Biotec |
| Anti-mouse CD24 Pacific Blue | M1/69 | Biolegend |
| Anti-mouse CD24 PE-Dazzle 594 | M1/69 | Biolegend |
| Anti-mouse CD25 AlexaFluor700 | PC61 | Biolegend |
| Anti-mouse CD69 PE/Cyanine5 | H1.2F3 | Thermo Fisher |
| Anti-mouse CD70 PerCP-eFluor710 | FR70 | Thermo Fisher |
| Anti-mouse H2Kb PerCP-eFluor 710 | AF6-88.5.5.3 | Thermo Fisher |
| Anti-mouse H2Kd PE | SF1-1.1.1 | Thermo Fisher |
| Anti-mouse ICOS VioGreen | REA192 | Miltenyi Biotec |
| Anti-mouse ICOSL PE | HK5.3 | Biolegend |
| Anti-mouse PD-1 APC | 29F.1A12 | Biolegend |
| Anti-mouse PD-L1 PE/Dazzle594 | 10F.9G2 | Biolegend |
| Anti-mouse PIR-B APC | 10-1-PIR | Thermo Fisher |
| Anti-mouse TIM-3 PE | REA602 | Miltenyi Biotec |

## Mixed leukocyte reaction (MLR)

Splenic pan-DCs were isolated from the spleens of untreated or 2.43 mAb-treated BALB/c mice as described above. Some DCs were set aside to confirm CD8α+ cDC1 depletion by flow cytometry. Allogeneic T-cells were isolated from the spleens of naïve C57BL/6 mice using the Pan T-cell isolation kit II (Miltenyi Biotec). Purified T-cells were stained with CellTrace Violet (Invitrogen). Splenic DCs were co-cultured with allogeneic T-cells in proliferation assay media (**PAM**) (RPMI-1640, 10% FBS, 0.5% Sodium Pyruvate, 0.5% MEM, and 55 μM BME) at a ratio of 1:5 and incubated at 37˚C with 7.5% $CO_2$. T-cells were stimulated with CD3/CD28 Dyna-Beads (Thermo Fisher Scientific) as a positive control. After 16–24 hours, rIL-2 (PeproTech) was added to each well at a final concentration of 50 IU/mL. Flow cytometry was performed to determine DC composition on day 2 of co-culture. After 3–4 days of co-culture flow cytometry was performed. Data were analyzed using Modfit software (Verity Software House) to determine the proliferation index (**PI**) of $H2K^b$+ T-cells.

## Tumor-killing assay

A BALB/c B-cell lymphoblastic leukemia cell line, A20-luciferase (A20-Luc), was provided by Dr. Xue-Zhong Yu, MD (Medical University of South Carolina) and has previously been used in murine BMT studies [20]. A20-luc was cultured in RPMI 1640 (Hyclone SH30027) with 10% FBS, MEM, sodium pyruvate (Hyclone SH30239), and 2 μL/mL Zeocin (Gibco) at 37 ˚C and 5% $CO_2$. This tumor-killing assay was performed according to previous reports [24] and optimized for our purposes. Splenic pan-DCs were isolated from the spleens of naïve or 2.43 mAb-treated BALB/c mice as described. Allogeneic T-cells were isolated from the spleens of naïve C57BL/6 mice using the Pan T-cell isolation kit II (Miltenyi Biotec). Splenic DCs were co-cultured with allogeneic T-cells in a 96-well U-bottom TC-treated plate at a ratio of 1:5 and incubated at 37˚C with 7.5% $CO_2$.

On day 3 of co-culture $1x10^5$, $2x10^5$, or $4x10^5$ A20-Luc mouse ($H2K^d$+) B-cell lymphoma target cells were added to the respective wells and mixed well with the pipette. Tumor alone controls were added to wells containing a volume of media equivalent to that of the DC:T-cell cultures. The following day, on day 4, each individual well was harvested and transferred into a black, clear-bottom, flat-bottom 96-well plate. Immediately prior to imaging, 50 μL of luciferin (GoldBio, LUCK) was added to all wells using a multi-channel pipette. Bioluminescence (**BLI**) was measured using LagoX (Special Instruments Imaging, Tucson, AZ) and quantified using Aura software (Verity). BLI was averaged for technical replicates. Specific killing was calculated as a percent of the tumor alone control at each respective tumor concentration.

## FACS cell sorting

Splenic pan DCs were isolated from nine age-matched, female, littermate BALB/c mice and pooled per three mice for a total of three biological replicates. Pooled DCs were washed in flow buffer (PBS with 0.5% FBS), incubated with anti-mouse Fc Block (Thermo Fisher Scientific), and then stained with CD8α PE-Cy7 (Thermo Fisher; clone: 53–6.7), and CD24 Pacific Blue (Biolegend; clone: M1/69). CD8α+ and pre-cDC1 ($CD24^{high}$ CD8α-) were sorted using FACSAria III (BD).

## RNA sequencing

All cell samples were stored in PBS and RNAlater (Thermo Fisher) at 4˚C prior to processing by the University of Arizona Genomics Core (UAGC). RNA Samples were assessed for quality with a High Sensitivity RNA Fragment Analyzer Kit (Advanced Analytics) and quantity with

an RNA HS assay kit (Qubit). Quality samples were used for library builds with the Rapid RNA Library Kit (Swift) and Dual Combinatorial Indexing Kit (Swift). Samples had quality and average fragment size assessed with the High Sensitivity NGS Analysis Kit (Advanced Analytics). Quantity was assessed with the Kapa Library Quantification kit (Illumina), and then samples were equimolar-pooled and clustered for sequencing using Illumina NextSeq500 run chemistry (NextSeq 500/550 High Output v2 kit 150 cycles). Data were sent to UAGC Bio-computing Group for further analysis.

## Informatics

Paired-end reads were demultiplexed using Illumina's BaseSpace service. Reads were trimmed using Trimmomatic version 0.32 (USADelLab). Fastq files were aligned to the GRCm38 version of the mouse reference genome using STAR version 2.5.2b [25]. Gene level quantification was performed using htseq-count version 0.6.1 [26]. DESeq2 was used to conduct differential expression analysis with genes having an adjusted p-value (padj) < 0.05 being considered significant [27]. Pathway enrichment was performed on the significant DEGs using QIAGEN's IPA software tool. P value = 0.05 corresponds to a 1.3 -log p-value and is considered statistically significant. Gene expression patterns of molecules for each identified pathway were used to determine a z-score, with a z-score >2 meaning "activated" and a z-score <-2 meaning "inhibited". RNA sequencing data have been deposited to the UA Research Data Repository reDATA and are publicly available (https://doi.org/10.25422/azu.data.14241902).

## Statistical analysis

Unpaired t-tests were used to determine significance between absolute counts, percent, MFI expression, and proliferation indices. Two-way ANOVA tests and Šidák's multiple comparisons tests were used to determine significance in CD8α+ cDC1 and pre-cDC1. Ordinary One-way ANOVA tests and Tukey's multiple comparisons tests were used to determine significance in percent specific killing of A20-Luc. P values <0.05 were considered statistically significant.

## Results

### Pre-cDC1s differ from CD8α+ cDC1s in their steady-state expression of co-stimulatory and co-inhibitory molecules

Splenic pan DCs were isolated from naïve mice and phenotypically characterized by flow cytometry. Pre-cDC1s are less abundant than CD8α+ cDC1s in the spleens of BALB/c mice (Fig 1A and 1B). Pre-cDC1s and CD8α+ cDC1s have been reported to express all the same PRRs and receptors for dead cell-associated Ag, and to have the same cross-presentation capabilities [10]. We first sought to determine the surface expression levels of various activating and inhibiting co-stimulatory molecules on the two DC subsets at steady state to confer potential differences in how they may stimulate adaptive T-cell responses. Compared to CD8α+ cDC1s, a lower percentage of pre-cDC1s express PD-L1 while those positively expressing PD-L1 have a higher MFI (Fig 1C). Notably, CD8+ cDC1s appear to have PD-L1$^{high}$ and PD-L1$^{dim}$ populations, whereas PD-L1-expressing pre-cDC1s primarily fall within the PD-L1$^{high}$ population, with relatively fewer PD-L1$^{dim}$ pre-cDC1s. Pre-cDC1s exhibit significantly greater percent expression of PIR-B (Fig 1D), a co-inhibitory molecule that suppresses lethal GvHD [28]. Regarding activating co-stimulatory molecules, pre-cDC1s express lower percent CD70 (Fig 1E), and greater percent ICOS-L (Fig 1F), with no differences in MFI. These data demonstrate that pre-cDC1s and CD8α+ cDC1s differ in their expression of various activating and inhibiting co-stimulatory molecules. While it is unclear whether the

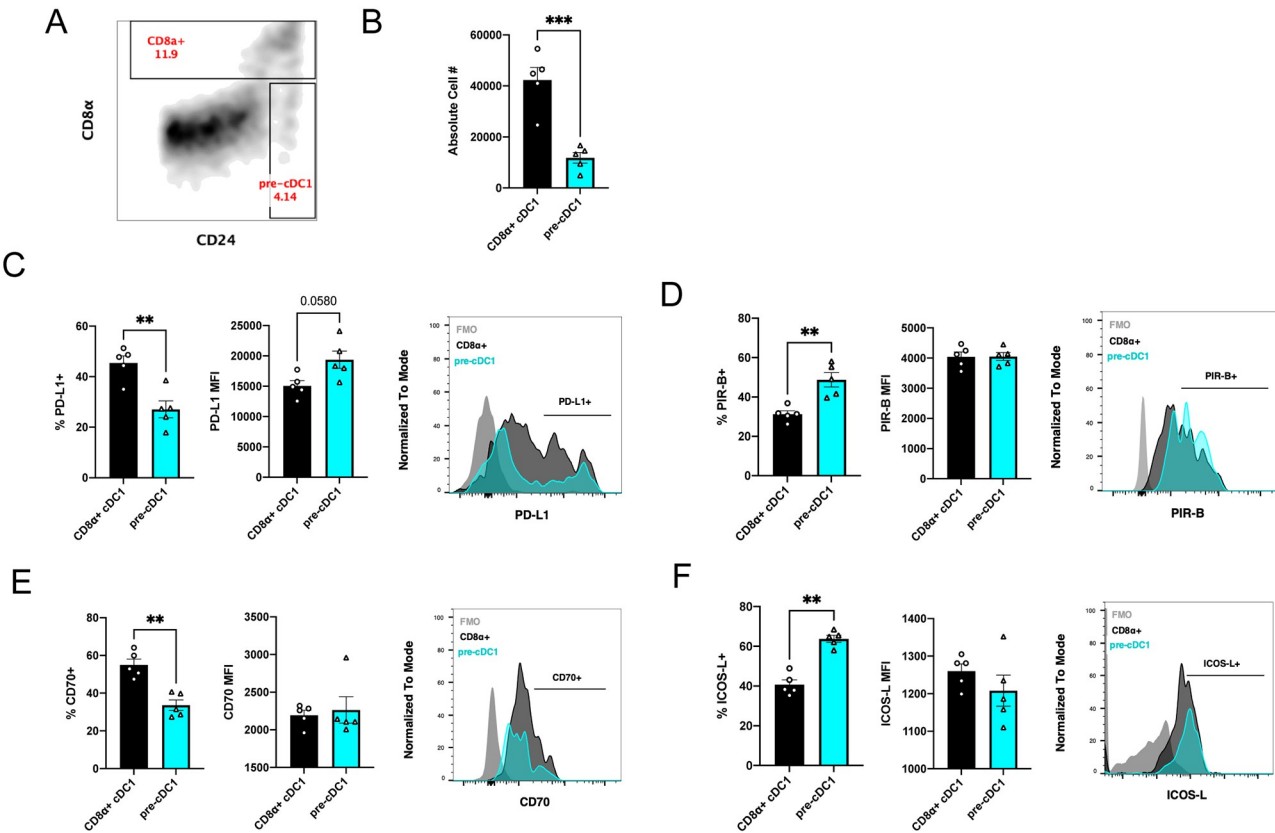

**Fig 1. Pre-cDC1s differ from CD8α+ cDC1s in their steady-state expression of co-stimulatory and co-inhibitory molecules.** Splenic pan DCs were isolated from naïve BALB/c mice and analyzed by flow cytometry to characterize pre-cDC1s and CD8α+ cDC1s. (A) Representative flow plot demonstrating gating strategy. Gated on CD11c+B220- (conventional DCs); pre-cDC1s are gated as CD24^highCD8α- and CD8α+ cDC1s are gated as CD8α+. (B) Mean absolute cell number of respective DC subsets shown with SEM. (C-F) Mean percent and MFI with representative histograms shown with SEM for expression of (C) PD-L1, (D) PIR-B, (E) CD70, and (F) ICOSL on CD8α+ cDC1s (black) and pre-cDC1s (cyan). Data is representative of two experiments. (n = 5 per group). Paired t tests were used to determine significance between DC subsets. **P<0.01, ***P<0.001.

expression pattern of pre-cDC1s results in a net function that is more immunogenic or tolerogenic compared to CD8α+ cDC1s, these results suggest potential differences in T-cell effector function modulation by these DC subsets following the same antigenic stimulus.

## Using a CD8α-depleting antibody to test functional redundancy

We next sought to compare the functions of pre-cDC1s and CD8α+ cDC1s. Unfortunately, there are no suitable mouse models commercially available that would allow us to properly investigate the separate functions of these DC subsets. The extreme rarity of these two DC populations complicate their examination *in vivo* and make cell-sorting for *ex vivo* assays unfeasible. Since the differentiating feature of these subsets is expression of CD8α, we used a CD8α-depleting antibody (2.43 mAb) to remove CD8α+ cDC1s *in vivo*. Depletion was confirmed by flow cytometry of splenocytes harvested from mice receiving CD8α-depleting antibody compared to untreated mice (Fig 2A and 2B). As previously reported, depletion of CD8α+ cDC1s also resulted in an increase in the proportion and absolute number of pre-cDC1s (Fig 2B and 2C) [10]. This strategy allowed us to evaluate splenic DC function in the presence and absence of CD8α+ cDC1s to determine whether pre-cDC1s exhibit functional redundancy. We therefore proceeded with the *ex vivo* proliferation assay outlined in the schematic in Fig 2D.

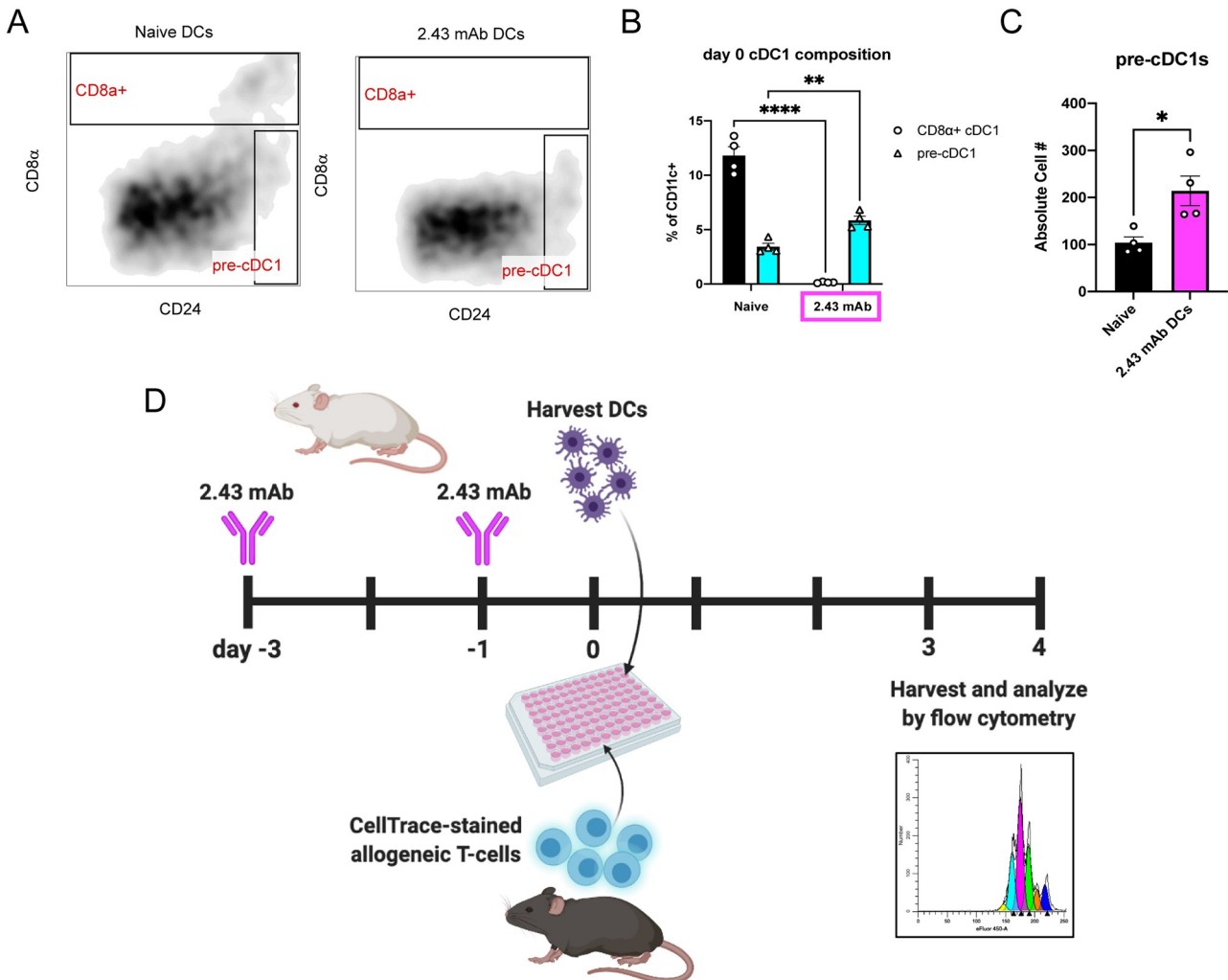

**Fig 2. Using a CD8α-depleting antibody to determine functional redundancy.** BALB/c mice were dosed with 200 μg of 2.43 mAb antibody intraperitoneally on day -3 and day -1. On day 0, splenic DCs were isolated and analyzed by flow cytometry. Data is representative of three independent experiments. (n = 4 per group) (A) Representative flow plots depicting relative cDC1 proportions in naïve DCs (left) and 2.43 mAb DCs (right). Gated on CD11c+B220- (conventional DCs). (B) Mean percent of CD8α+ cDC1s (black) and pre-cDC1s (cyan) in untreated and 2.43 mAb-treated DCs shown with SEM. Two-way ANOVA and Dunnett's multiple comparisons used to determine significance. $**P<0.01$, $****P<0.0001$. (C) Mean absolute cell number of pre-cDC1s (CD24$^{high}$CD8α-) in untreated (black) and 2.43 mAb-treated (magenta) groups shown with SEM. Unpaired t test used to determine significance. $*P<0.05$. (D) Schematic depicting experimental design for proposed allogeneic mixed leukocyte reaction (MLR) using 2.43 mAb treatment.

## 2.43 mAb-treated DCs induce less proliferation, altered PD-1 expression, and depressed induction of cell death of alloreactive T-cells

We first sought to compare the ability of 2.43 mAb-treated DCs to stimulate alloreactive T-cell proliferation using an *ex vivo* allogeneic MLR. Using ModFit software, we first gated on H2K$^b$+ cells to distinguish T-cells, then fit the parent peak using each corresponding T-cell alone control and generated the representative CellTrace dilution histograms for each experimental condition and timepoint (Fig 3A). ModFit software calculates a proliferation index (**PI**) based on the area under the curves representing consecutive cell divisions. We found that 2.43 mAb-treated DCs induced significantly less allogeneic T-cell proliferation on day 3 (Fig

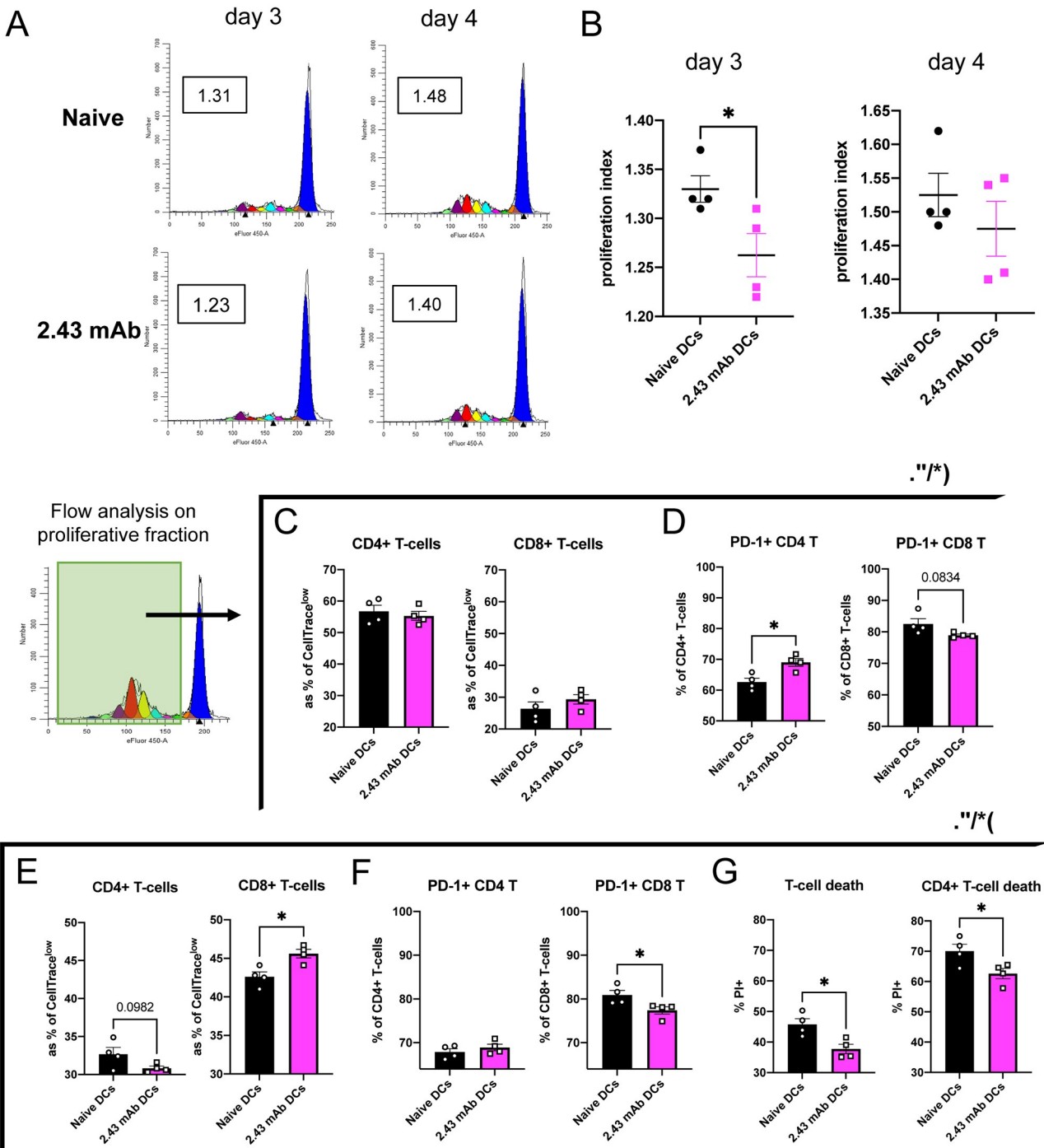

**Fig 3. 2.43 mAb-treated DCs induce less proliferation, altered PD-1 expression, and depressed induction of cell death of alloreactive T-cells.**
BALB/c mice were dosed with 200 μg of 2.43 mAb antibody intraperitoneally on day -3 and day -1. On day 0, splenic DCs were isolated and plated in an allogeneic MLR. Allogeneic T-cell proliferation was assessed on day 3 and day 4 of co-culture. Data is representative of two independent experiments. (n = 4 per group) (A-B) ModFit Software analysis of CellTrace dye dilutions. (A) Representative histograms depicting proliferation of H2K$^b$+ T-cells following stimulation with naïve DCs (top) or 2.43 mAb DCs (bottom) on days 3 and 4. Proliferation Index (PI) for representative histograms is boxed. (B) Mean PI for naïve or 2.43 mAb DCs shown with SEM. (C-G) Flow analysis of the proliferative fraction of T-cells was determined by first gating on H2K$^b$+CellTrace$^{low}$ T-cells. (C) Mean percent of proliferated T-cells that are CD4+ (left) and CD8+ (right) on day 3 shown with SEM. (D) Mean percent of proliferated CD4+ (left) and CD8+ (right) T-cells expressing PD-1 on day 3 shown with SEM. (E) Mean percent of proliferated T-cells that are CD4+ (left) and CD8+ (right) on day 4 shown with SEM. (F) Mean percent of proliferated CD4+ (left) and CD8+ (right) T-cells expressing PD-1 on day 4 shown with SEM. (G) Mean percent of proliferated T-cells (left) and CD4+ T-cells (right) that are Propidium Iodide+ shown with SEM. Unpaired t tests were used to determine significance between groups. *P<0.05.

3B), though by day 4 this was no longer significant. We also characterized the proliferative fraction of T-cells via flow cytometry by first gating on H2K$^b$+CellTrace$^{low}$ T-cells. On day 3 of the assay, we observed more CD4+ T-cell proliferation compared to CD8+ T-cell proliferation in both naïve and 2.43 mAb-treated conditions (Fig 3C). We further evaluated their expression of various surface markers of T-cell activation, exhaustion, and anergy. We found that on day 3, 2.43 mAb-treated DCs induced significantly greater expression of PD-1 on CD4+ T-cells and trended toward less expression on CD8+ T-cells, compared to untreated DCs (Fig 3D). Analysis of the proliferative fraction of T-cells was performed on day 4 of the assay and found that 2.43 mAb-treated DCs resulted in greater representation of CD8+ T-cells within the proliferative fraction (Fig 3E). We additionally found that among proliferated CD8+ T-cells, significantly fewer of them expressed PD-1 (Fig 3F). On day 4 we assessed induction of alloreactive T-cell death, a reported function of CD8α+ cDC1s, and found that 2.43 mAb-treated DCs induced significantly less alloreactive T-cell death compared to untreated DCs, and this was primarily driven by reduced induction of CD4+ T-cell death (Fig 3G) [18, 29].

## 2.43 mAb-treated DCs retain a lower frequency of CD8α+ cDC1s in *ex vivo* cultures

We next wanted to know how long CD8α+ cDC1 depletion persisted *ex vivo* since pre-cDC1s may mature into CD8α+ cDC1s upon entering the spleen and receiving tissue-specific signals. We therefore analyzed DCs in untreated versus 2.43 mAb-treated conditions on day 2 of *ex vivo* co-culture with allogeneic T-cells, gating on H2K$^d$+CD11c+ cells to distinguish them (Fig 4A) and found a significantly lower frequency of H2K$^d$+CD11c+ DCs present in cultures on day 2 under 2.43 mAb-treated conditions (Fig 4B). We further assessed the composition of cDC1s as a percent of H2K$^d$+CD11c+ DCs (Fig 4C) and confirmed persistence of a significantly lower proportion of CD8α+ cDC1s in the 2.43 mAb-treated group compared to untreated (Fig 4D). These confounding factors of overall fewer DCs in culture and reduced proportion of CD8α+ cDC1s may contribute to the differential effects on T-cell proliferation, PD-1 expression, and induced cell death.

## Depletion of CD8α+ cDC1s results in improved tumor-killing

We additionally sought to evaluate the ability of 2.43 mAb-treated DCs to induce anti-tumor responses. We plated naïve and 2.43 mAb DCs with allogeneic T-cells on day 0, added A20 mouse lymphoma cells transduced to express luciferase (**A20-Luc**) on day 3 of co-culture, and then measured bioluminescence (**BLI**) of A20-Luc tumor cells on day 4 (Fig 5A) [24]. A representative plate image depicts the BLI readout (Fig 5B). Tumor cytotoxicity was calculated as a percent of the tumor alone BLI for each tumor concentration. Our results demonstrate that 2.43 mAb-treated DCs resulted in induction of significantly higher killing of A20-Luc tumor cells compared to untreated DCs at tumor concentrations of 1x10$^5$ (Fig 5C) and 2x10$^5$ (Fig 5D). These data indicate that the depletion of CD8α+ cDC1s with 2.43 mAb can enhance DC-induced anti-leukemic cytotoxic activity despite having less alloreactive T-cell proliferation.

## Pre-cDC1 and CD8α+ cDC1 exhibit distinct transcriptional programs

We finally wanted to determine whether murine pre-cDC1s and CD8α+ cDC1s differ transcriptionally. Bulk RNA sequencing of cell-sorted pre-cDC1s and CD8α+ cDC1s was performed with the help of the University of Arizona Genomics Core (UAGC). These data revealed significant differences in the expression of various transcripts, exemplified in a volcano plot (Fig 6A). Among the most significantly downregulated genes in pre-cDC1s compared to mature CD8α+ cDC1s are Tnfrsf4 (OX40), IL-12p40, and Cxcl11. OX40 expression

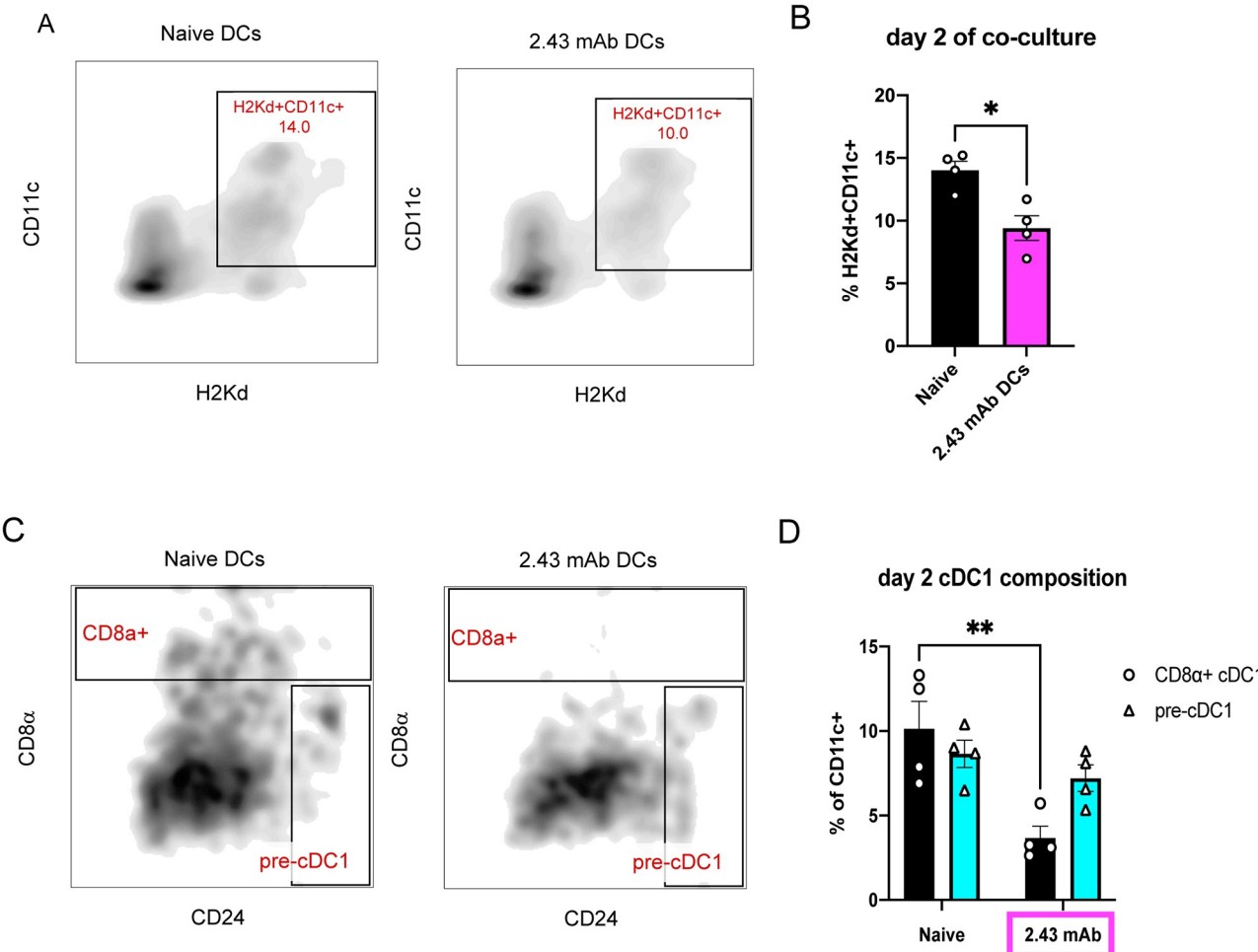

**Fig 4. 2.43 mAb-treated DCs retain a lower frequency of CD8α+ cDC1s.** BALB/c mice were dosed with 200 μg of 2.43 mAb antibody intraperitoneally on day -3 and day -1. On day 0, splenic DCs were isolated and plated in an allogeneic MLR. DC composition was determined by flow cytometry on day 2 of co-culture. Data is representative of two independent experiments. (n = 4 per group) (A) Representative flow plots depicting percent of H2K$^d$+CD11c+ DCs in naïve DCs (left) and 2.43 mAb DCs (right) on day 2 of co-culture. (B) Mean percent of H2K$^d$+CD11c+ DCs for naïve (black) or 2.43 mAb DCs (magenta) shown with SEM. Unpaired t test was used to determine significance between groups. $^{**}$P<0.01. (C) Representative flow plots depicting percent CD8α+ cDC1s and pre-cDC1s in naïve (left) and 2.43 mAb DCs (right). (D) Mean percent CD8α+ cDC1s (black bars) and pre-cDC1s (teal bars) in naïve and 2.43 mAb DCs shown with SEM. 2way ANOVA and Šidák's multiple comparison used to determine statistical significance. $^{**}$P<0.01.

on pDCs has been shown to promote anti-tumor immunity, however, to our knowledge, expression of OX40 on CD8α+ cDC1s has not been previously reported [30]. IL-12p40 and Cxcl11 are pro-inflammatory and support the recruitment of lymphocytes, so their downregulation may suggest that pre-cDC1 are less pro-inflammatory. Among the most significantly upregulated genes in pre-cDC1 are Dram1, Dstn, and several Ig variable transcripts thought to be involved in antigen binding. Dram1 (DNA-damage-regulated autophagy modulator) regulates autophagy in part through effects on lysosomes and plays an important role in inducing selective autophagy in response to recognition of intracellular pathogens [31–33]. Dstn (Destrin) is a depolymerizing factor that enhances the turnover rate of actin, which plays a significant role in the capture and presentation of antigen [34].

Additionally, we find that pre-cDC1s differentially regulate several genes that are involved in Ag processing and presentation pathways. Igkv7-33, Ighv1-12, and Igkv1-99 were all

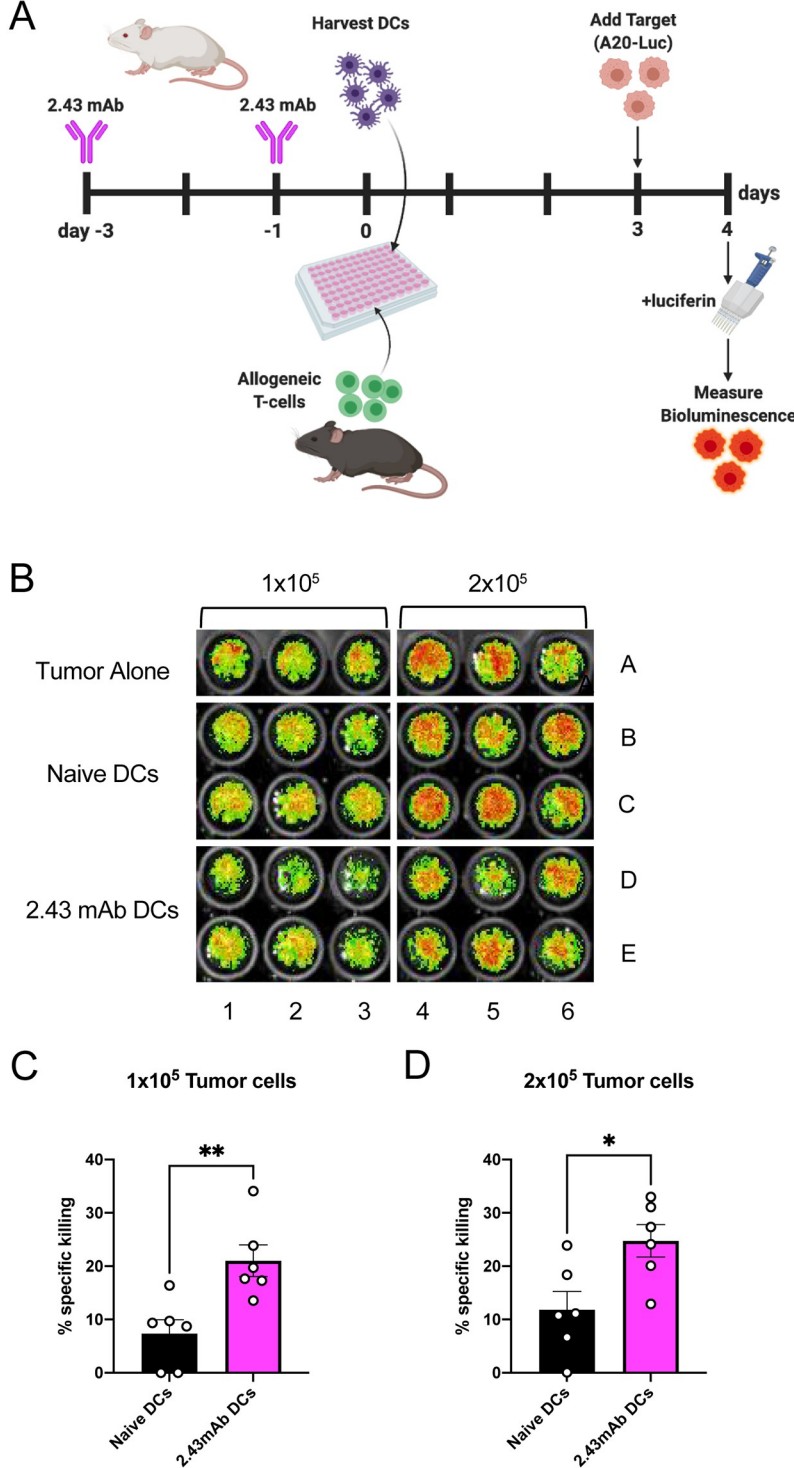

**Fig 5. Depletion of CD8α+ cDC1s results in improved tumor-killing.** BALB/c mice were dosed with 200 μg of 2.43 mAb antibody intraperitoneally on day -3 and day -1. On day 0, splenic DCs were isolated and plated with allogeneic T-cells. On day 3 of co-culture A20-Luc tumor cells were added to wells at the indicated concentrations. On day 4, luciferin was added to wells and plates were imaged. BLI was measured for each independent well and averaged among technical replicates. BLI values were used to quantify specific killing as a percent of tumor alone control. Data is pooled from two independent experiments. (n = 6 per group) (A) Schematic depicting the experimental timeline. (B) Representative bioluminescence image depicting plate layout and showing BLI intensity on a scale from low (green) to high (red) indicative tumor burden. (C-D) Mean percent specific killing is shown with SEM for A20-Luc

concentrations of (C) $1x10^5$ and (D) $2x10^5$ A20-Luc tumor cells. Unpaired t test used to determine significance between groups. $^*P<0.05$, $^{**}P<0.01$.

significantly upregulated in pre-cDC1s and have vague roles in antigen binding and innate immune responses. Dstn (Destrin), an actin depolymerizing factor, was upregulated and contributes to the efficient capture and presentation of Ag [34]. Dram1 was upregulated and is associated with selective induction of autophagy [31]. Based on IPA results, mTOR signaling is predicted to be inhibited in pre-cDC1s compared to CD8α+ cDC1s, which has been linked to cytoprotective induction of autophagy [35]. Altogether, the RNAseq results suggest significant differences in how Ag processing and presentation, particularly with respect to autophagy, is regulated in pre-cDC1s compared to CD8α+ cDC1s at steady state.

Ingenuity pathway analysis (IPA) revealed a number of pathways predicted to be inhibited in pre-cDC1 compared to CD8α+ cDC1 including ICOS-ICOSL signaling, DC Maturation, and CD40 signaling (Fig 6B). These findings are in line with the activation state expected of immature, precursors to DCs [36]. Interestingly, we find that Flt3 signaling is also inhibited in pre-cDC1s compared to CD8α+ cDC1s, though it has been recently reported that Flt3L is dispensable for the final maturation step of cDC1s [37]. Pathways that were revealed to be activated in pre-cDC1s included several pathways regulating cell cycle control, oxidative phosphorylation, and the PD-1/PD-L1 cancer immunotherapy pathway (Fig 6C). Differentially activated cell cycle control may reflect the difference in lifespan of cDC1s compared to CD8α+ cDC1s [10]. A metabolic program favoring oxidative phosphorylation has been associated with tolerogenic DCs [38]. Predicted activation of the PD-1/PD-L1 pathway is interesting because we found that a smaller percentage of pre-cDC1s express PD-L1 compared to CD8α+ cDC1, but those pre-cDC1s that do positively express PD-L1 appear to express higher levels of it (Fig 1C). A heat map was generated for the PD-1/PD-L1 pathway and displays very clear differences in gene expression levels between pre-cDC1s and CD8α+ cDC1s (Fig 6D).

## Discussion

The function of pre-cDC1s has not been extensively studied, in part because they have only recently been identified as the committed precursor to CD8α+ cDC1s [1, 4]. The few mechanistic reports that have directly compared the two DC subsets have determined that pre-cDC1s exhibit a significantly enhanced lifespan and induce greater expansion of Ag-specific memory CD8+ T-cells and control of viral load [5, 10]. There are still many gaps in our knowledge of how pre-cDC1s function as compared to CD8α+ cDC1s, as well as what factors trigger their immigration from the bone marrow to the spleen and maturation into CD8α+ cDC1s. Given the crucial role of CD8α+ cDC1s in suppressing GvHD while promoting GvL responses, the role of pre-cDC1s in induction of alloreactive T-cell responses and anti-tumor responses is of great clinical significance for HCT.

Our findings indicate that pre-cDC1s express significantly different levels of various co-signaling molecules compared to CD8α+ cDC1s. We observed that a significantly higher percentage of pre-cDC1s express PIR-B, which has been shown to regulate cytotoxic T-cell responses and prevent lethal GvHD, which would be advantageous in the HCT setting [27, 38, 39]. We also found that a lower percentage express PD-L1, which provides inhibitory signals to T-cells that can help mitigate GvHD [40–42]. Notably, CD8α+ cDC1s appear to have both PD-L1^high and PD-L1^dim populations, whereas positively expressing pre-cDC1s primarily fall into the PD-L1^high population. We additionally documented a significantly lower percentage of pre-cDC1s expressing CD70 compared to CD8α+ cDC1s. CD70 activates CD27 on T-cells which plays a nonredundant role in regulating T-cell, B-cell, and NK cell activity [43]. Studies have

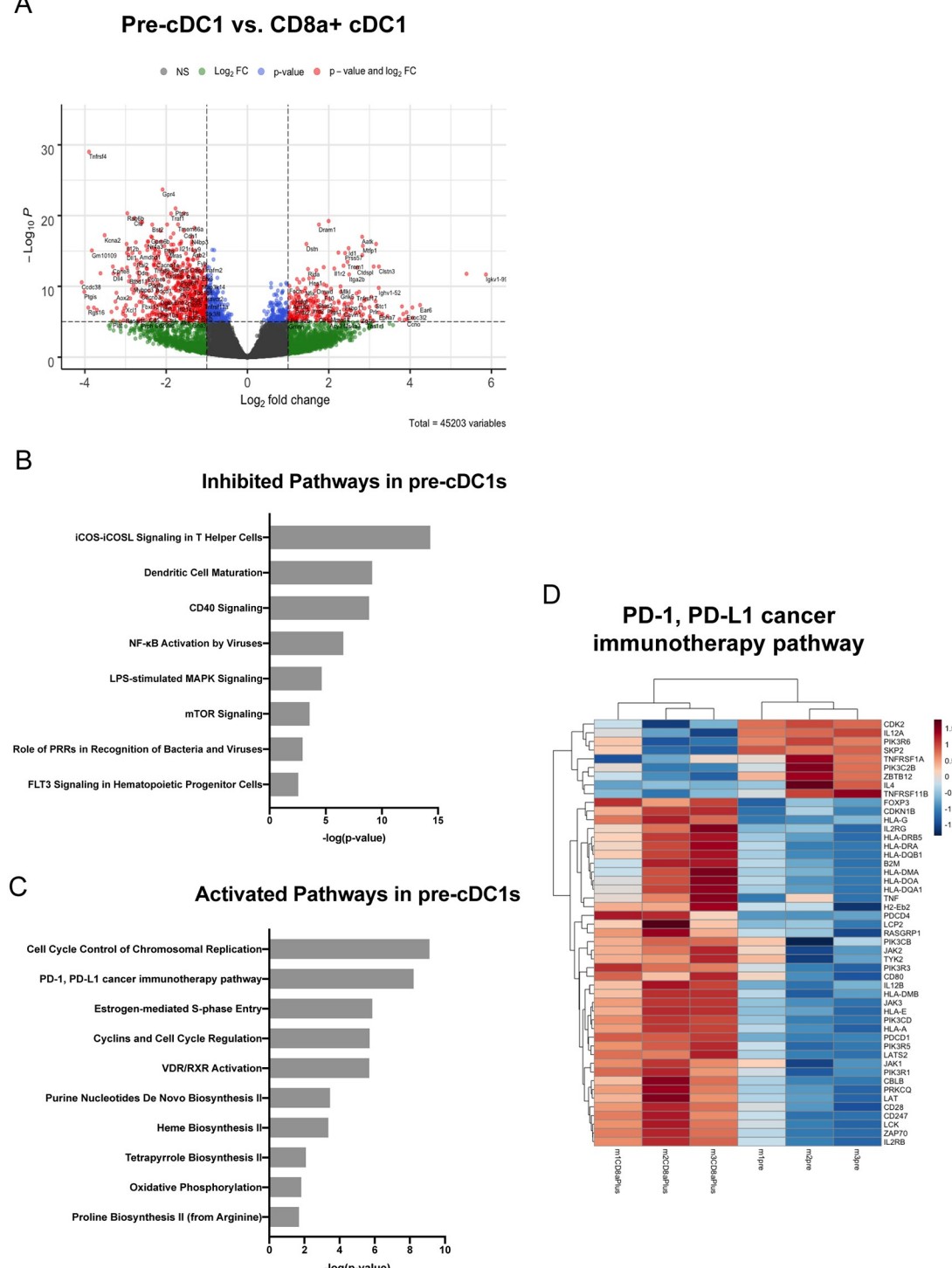

**Fig 6. Pre-cDC1 and CD8α+ cDC1 exhibit distinct transcriptional programs.** Splenic DCs were isolated from naïve BALB/c mice and pre-cDC1s and CD8α+ cDC1s were cell sorted for RNA sequencing. (A) Enhanced volcano plot identifying genes that are up or down-regulated in pre-cDC1s compared to CD8α+ cDC1s. (B-C) Pathway results using Qiagen's IPA software. P = 0.05 corresponds to a 1.3 -log p-value, and anything above this value is considered statistically significant. (B) Pathways predicted to be significantly inhibited (z-score < -2) in pre-cDC1s compared to CD8α+ cDC1s. (C) Pathways predicted to be significantly activated (z-score < 2) in pre-cDC1s compared to CD8α+ cDC1s. (D) Heat map showing relative gene expression levels for the PD-1/PD-L1 cancer immunotherapy pathway. Full documentation, methodology, and raw data for this data set is available online at: https://doi.org/10.25422/azu.data.14241902.

indicated that CD70 may prevent the induction of T-cell tolerance by steady state DCs, suggesting that lower expression by pre-cDC1s may be advantageous in HCT by promoting tolerance [44]. We also report that a significantly greater percentage of pre-cDC1s express ICOSL, which contributes to activation of T-cells and would therefore exacerbate GvHD [45]. This finding contradicts our RNAseq results which indicate that the ICOS-ICOSL signaling in T helper cells pathway is significantly inhibited in pre-cDC1s compared to CD8α+ cDC1s. Overall, these findings make it difficult to clearly ascertain the net effect of pre-cDC1s in inducing or protecting from GvHD when compared to CD8α+ cDC1s. A future line of work employing ICOS and ICOSL blockades would be helpful in delineating the role of this co-signaling molecule in the overall function of pre-cDC1s.

No mouse models currently exist that would allow us to adequately investigate the respective roles of pre-cDC1s and CD8α+ cDC1s without resulting in significant changes to other immune cell subsets. The extreme rarity of these DC subsets further complicates their investigation *in vivo*. We therefore examined the function of splenic pan-DCs *ex vivo* following *in vivo* treatment with an anti-CD8α monoclonal antibody to deplete CD8α+ cDC1s, thereby indirectly assessing the role of CD8α+ cDC1s compared to pre-cDC1s. This imperfect approach for distinguishing the relative functions of pre-cDC1s and CD8α+ cDC1s has several caveats worth mentioning. First, in using a CD8α-depleting antibody, we are depleting CD8 + T-cells in addition to CD8α+ cDC1s, which may have contributing effects on spleen cells and DCs specifically. Secondly, as others have shown, we find that depletion of CD8α+ cDC1s results in an increase in the percent and absolute number of pre-cDC1s in the spleen [5]. This was expected as we know that pre-cDC1s seed peripheral tissues from the BM in order to replenish mature CD8α+ cDC1s, however it may have unknown effects on splenic DC function. Newly seeded pre-cDC1s may behave differently compared to pre-cDC1s that have resided in the spleen longer. For instance, it has been demonstrated that pre-cDCs in the BM express very high levels of Ag processing enzymes and downregulate their expression upon migrating into peripheral tissues [17].

Allogeneic MLR results demonstrated that 2.43 mAb-treated DCs induce significantly less alloreactive T-cell proliferation *ex vivo* with a slight preference for proliferation of CD8+ T-cells over CD4+ T-cells. Moreover, 2.43 mAb-treated DCs resulted in greater expression of PD-1 on CD4+ T-cells, lower expression of PD-1 on CD8+ T-cells, as well as reduced cell death of CD4+ T-cells. Interpretation of these results is limited by the fact that 2.43 mAb depletion resulted in a lower percentage of H2K$^d$+CD11c+ cells on day 2 of culture in the MLR assays. The reason for this is unclear but suggests that 2.43 mAb-treated DCs did not persist as well *ex vivo* compared to untreated DCs, which would contribute to their observed decrease in inducing allogeneic T-cell proliferation. It is also possible that 2.43 mAb-treated DCs may be downregulating H2K$^d$ or CD11c expression compared to untreated DCs. Nevertheless, evaluation of cDC composition of H2K$^d$+CD11c+ cells on day 2 of *ex vivo* culture demonstrated that 2.43 mAb-treated DCs maintained a predominant pre-cDC1 proportion over CD8α+ cDC1. Thus, we are confident that the differential T-cell responses observed are indicative of the relative abundance of pre-cDC1s and CD8α+ cDC1s.

Tumor killing assays revealed enhanced cytotoxicity against A20 B-cell lymphoblastic leukemia cell line induced by 2.43 mAb-treated DCs compared to untreated DCs, despite also having decreased stimulatory capacity of allogeneic T-cells. These findings suggest that 2.43 mAb-treated DCs may be skewed toward favoring GvL over GvHD. One would expect less alloreactive T-cell proliferation to translate to less tumor killing, though we cannot confirm from these studies whether or not tumor killing is T-cell-mediated. We also cannot rule out the possibility that 2.43 mAb-treated DCs are themselves cytotoxic. Reviewed by Larmonier *et. al.* and Hanke *et. al.*, DCs have been found to function as direct cytotoxic effectors against

tumor cells [46, 47]. This less conventional function of DCs has been found to be highly dependent upon NO synthase expression, peroxynitrites, and iNOS expression [48, 49]. Cytotoxic DC function has also been shown to be differentially regulated by PIAS-1 and STAT3 [50]. Furthermore, human DCs have also been shown to have cytotoxic function, also dependent upon peroxynitrites and restricted to immature DCs [51]. Future work is needed to investigate these mechanisms of cytotoxic function specifically in pre-cDC1s to determine whether they truly function as killer DCs.

RNAseq results comparing cell sorted pre-cDC1s and CD8α+ cDC1s reveal significant differences in the transcriptional programs of these DC subsets that have otherwise been thought of as having similar, if not identical, functions. Among the pathways predicted to be inhibited in pre-cDC1s include DC maturation, CD40 signaling, NFkB activation, LPS-stimulated MAPK signaling, and the role of PRRs in recognition of bacteria and viruses. Overall, inhibition of these pathways is indicative an immature DC phenotype, which is unsurprising for a precursor DC but may also indicate a proclivity to induce tolerogenic responses. Additionally, the vitamin D VDR/RXR activation and oxidative phosphorylation pathways are predicted to be activated in pre-cDC1s and have both been linked to tolerogenic DC responses [38, 52]. The PD-1/PD-L1 is also predicted to be activated in pre-cDC1s compared to CD8α+ cDC1s, which would be highly beneficial for induction of tolerance in the HCT setting. Interestingly, while PD-1 signaling is critical to induction and maintenance of peripheral tolerance in transplantation, the induction of programmed cell death of alloreactive T-cells has been specifically linked to PD-L1, and its ligand is not believed to be PD-1 [53–57].

Given the more recent identification of pre-cDC1s, this DC subset has not been extensively studied in the contexts of alloreactivity and anti-tumor responses and genetically modified mouse models for investigating this rare cell type have not yet been developed. Limitations in our overall methodology make it difficult to make conclusions regarding the ability of pre-cDC1s to control allogeneic T-cell responses *in vivo*. Though, it is worth nothing that previous work from our laboratory has found a strong association between host pre-cDC1 abundance and improved survival from lethal GvHD in a major-mismatched bone marrow transplantation model [22]. Nevertheless, tumor-killing assays suggest that pre-cDC1s may support a vigorous cytotoxic response against leukemic cells *ex vivo*, particularly in the presence of limited Ag. Although a human equivalent of pre-cDC1s has not yet been identified, our understanding of human DC biology is changing very rapidly with sequencing technology [58–60]. Obtained RNA sequencing results suggest that these two DC populations are transcriptionally distinct. Given the highly conserved features of cDC1s between mice and humans, and the recent discovery of new DC subsets found in both species, it is plausible that humans express a "pre-cDC1" population as well [59, 61, 62].

In summary, pre-cDC1s exhibit significantly different expression patterns of various co-stimulatory molecules compared to CD8α+ cDC1s, suggesting that pre-cDC1s may induce different T effector responses than CD8α+ cDC1s when given the same antigenic stimulus. Our results also indicate that, in the absence of CD8α+ cDC1s, pre-cDC1s induce a more potent anti-tumor response in the presence of limited Ag. However, we may only speculate as to how pre-cDC1s function *in vivo*, and a better system is needed to more confidently and extensively investigate the role of pre-cDC1s in alloreactivity and anti-tumor responses. RNAseq results suggest an immature, tolerogenic function of pre-cDC1s and differential regulation of Ag processing and cross-presentation machinery. Altogether, these findings indicate a unique function of pre-cDC1s that warrants further investigation and consideration in future studies of anti-cancer responses and HCT using murine models.

## Acknowledgments

The authors wish to thank Vanessa Frisinger for administrative assistance. We'd also like to thank the University of Arizona Flow Cytometry Core Facility for the use of their analytical software and facilities, as well as the University of Arizona Genomics Core Facility for processing and generation of RNAseq data and informatics expertise.

## Author Contributions

**Conceptualization:** Megan S. Molina, Emely A. Hoffman, Jessica Stokes, Richard J. Simpson, Emmanuel Katsanis.

**Data curation:** Megan S. Molina, Emely A. Hoffman, Jessica Stokes, Nicole Kummet.

**Formal analysis:** Megan S. Molina.

**Funding acquisition:** Emmanuel Katsanis.

**Investigation:** Megan S. Molina.

**Methodology:** Megan S. Molina, Emely A. Hoffman, Jessica Stokes, Nicole Kummet.

**Supervision:** Emmanuel Katsanis.

**Writing – original draft:** Megan S. Molina.

**Writing – review & editing:** Megan S. Molina, Emely A. Hoffman, Jessica Stokes, Richard J. Simpson, Emmanuel Katsanis.

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
