## [Decision Letter · Decision Letter 0]

26 Apr 2022

PONE-D-21-26682Murine Precursors to Type 1 Conventional Dendritic Cells Induce Tumor Cytotoxicity and Exhibit Activated PD-1/PD-L1 PathwayPLOS ONE

Dear Dr. Molina,

Thank you for submitting your manuscript to PLOS ONE. After careful consideration, we feel that it has merit but does not fully meet PLOS ONE’s publication criteria as it currently stands. Therefore, we invite you to submit a revised version of the manuscript that addresses the points raised during the review process.

We have now received reports from two referees of your manuscript, as agree with reviewers comments raised a few concerns about this study. After careful consideration, we invite you to submit a revised version of the manuscript which correlate the in vitro and in vivo data. 

We look forward to receiving your revised manuscript.

Kind regards,

Senthilnathan Palaniyandi, Ph.D

Academic Editor

PLOS ONE

https://journals.plos.org/plosone/s/file?id=ba62/PLOSOne_formatting_sample_title_authors_affiliations.pdf.2.

2. In your Methods section, please provide additional information on the animal research and ensure you have included details on : (1) methods of sacrifice (2) methods of anesthesia and/or analgesia, and (2) efforts to alleviate suffering.

3. We note that you are reporting an analysis of a microarray, next-generation sequencing, or deep sequencing data set. PLOS requires that authors comply with field-specific standards for preparation, recording, and deposition of data in repositories appropriate to their field. Please upload these data to a stable, public repository (such as ArrayExpress, Gene Expression Omnibus (GEO), DNA Data Bank of Japan (DDBJ), NCBI GenBank, NCBI Sequence Read Archive, or EMBL Nucleotide Sequence Database (ENA)). In your revised cover letter, please provide the relevant accession numbers that may be used to access these data. For a full list of recommended repositories, see http://journals.plos.org/plosone/s/data-availability#loc-omics or http://journals.plos.org/plosone/s/data-availability#loc-sequencing.

We will update your Data Availability statement on your behalf to reflect the information you provide."

Reviewers' comments:

Reviewer's Responses to Questions

**Comments to the Author**

1. Is the manuscript technically sound, and do the data support the conclusions?

Reviewer #1: No

Reviewer #2: Yes

2. Has the statistical analysis been performed appropriately and rigorously? 

Reviewer #1: Yes

Reviewer #2: Yes

3. Have the authors made all data underlying the findings in their manuscript fully available?

Reviewer #1: Yes

Reviewer #2: Yes

4. Is the manuscript presented in an intelligible fashion and written in standard English?

Reviewer #1: Yes

Reviewer #2: Yes

5. Review Comments to the Author

Reviewer #1: Mature CD8α+ cDC1s are recognized for suppressing graft-versus-host disease (GvHD) while promoting graft-versus-leukemia (GvL), however pre-cDC1s have not previously been investigated in the context of alloreactivity or anti-tumor responses. The authors have characterized pre-cDC1s, compared to CD8α+ cDC1s, found that a lower percentage of pre- cDC1s express PD-L1, yet express greater PD-L1 by MFI and a greater percent PIR-B, a GvHD- suppressing molecule. Functional assays were performed in vitro following in vivo depletion of CD8α+ DCs to examine whether pre-cDC1s play a redundant role in alloreactivity. Bulk RNA sequencing revealed distinct transcriptional programs of these DCs, with pre-cDC1s exhibiting activated PD-1/PD-L1 signaling compared to CD8α+ cDC1s. Thesresults indicate distinct T-cell-priming capabilities of murine pre-cDC1s compared to CD8α+ cDC1s with highly clinically relevant implications in suppressing GvHD while promoting GvL responses, highlighting the need for greater investigation of murine pre-cDC1s.

The in vitro data follows the logic of this manuscript, unfortunately there are no in vivo data to suggest that what is observed in vitro has an impact in an in vivo model. The use of MLR type of assays and in vitro GVL studies are not necessarily representative of what happens in vivo. Without an in vivo analysis, these data are not likely to be definitive enough for the conclusions drawn by the investigators.

Reviewer #2: I don't find any issues that need to be addressed by the Authors. The study does provide important new findings in the field and in my opinion, the manuscript can be accepted and published at its current state.

6. PLOS authors have the option to publish the peer review history of their article (what does this mean?). If published, this will include your full peer review and any attached files.

Reviewer #1: No

Reviewer #2: No

---

## [Author Response · Author response to Decision Letter 0]

17 May 2022

Dear Dr. Palaniyandi,

Please find attached our revised manuscript entitled “Murine Precursors to Type 1 Conventional Dendritic Cells Induce Tumor Cytotoxicity and Exhibit Activated PD-1/PD-L1 Pathway” submitted to PLOS ONE (PONE-D-21-26682). We are very pleased that our manuscript was deemed acceptable upon completion of minor revisions requested by one of the two reviewers. We thank the reviewer for their thoughtful comments which have been addressed as detailed below and in the revised manuscript. All changes to the manuscript text are highlighted by track changes.

Journal Requirements:

A number of journal-specific requirements regarding manuscript style and format, data availability, and details on animal research methods and ethics were addressed in our manuscript. These changes are detailed below:

1. We have edited our manuscript to meet PLOS ONE’s style requirements, per the PLOS ONE formatting guide provided by the editor.

2. Our Methods section, subsection “Mice”, was revised to include details on the method of sacrifice (Line 106-108).

3. Our RNA sequencing data set has been uploaded to a public repository in compliance with PLOS ONE’s guidelines. RNA sequencing data have been deposited to the UA Research Data Repository reDATA and are publicly available at the following DOI: https://doi.org/10.25422/azu.data.14241902

4. Our manuscript does not contain any blot or gel results that pertain to PLOS ONE’s policy on blot/gel reporting and figure preparation.

5. In regard to PLOS ONE’s data availability policy:

a. There are no ethical or legal restrictions on sharing the data sets used within our manuscript.

b. RNA sequencing data have been deposited to the UA Research Data Repository reDATA and are publicly available at the following DOI: https://doi.org/10.25422/azu.data.14241902

6. We have removed all instances of “data not shown” within our manuscript (Line 292 and Line 308) to comply with PLOS ONE’s policies on data accessibility, as these data were deemed impertinent to the conclusions drawn from our study.

7. We have included our full ethics statement within the Methods section (Lines 96-100).

Reviewer Comments:

Reviewer #1 stated that the in vitro data follows the logic of our manuscript, however there are no in vivo data suggest that our in vitro findings are relevant to what happens in vivo. They further state that our MLR and tumor-killing assays are not necessarily representative of what happens in vivo and that without further in vivo analysis, our data are not definitive enough for the conclusions drawn.

1. We agree with Reviewer #1 that performing in vivo experiments would provide more compelling data and better portray the biological impact of pre-cDC1 function compared to that of CD8α+ cDC1s. Unfortunately, we are limited in our ability to perform rigorous investigations of these DC subsets in vivo, as noted in our discussion (Lines 639-642). Limiting factors include:

a. Lack of suitable mouse models: While Batf3 KO mice are a well-established mouse model that lack CD8α+ cDC1s and retain pre-cDC1s, these mice exhibit significant deficiencies and overall changes in the DC compartment. As such, their ability to mount adaptive T-cell responses is severely altered making this mouse system unsuitable for addressing the questions at hand.

b. Extreme rarity: pre-cDC1s and CD8α+ cDC1s are two extremely rare cell subsets, with <50,000 CD8α+ cDC1s and <20,000 pre-cDC1s isolated from the spleens of naïve mice. These exceedingly low cell numbers make it extremely technically challenging to evaluate pre-cDC1 and CD8α+ cDC1 function in vivo. In addition, mouse models of bone marrow transplantation require whole body irradiation and chemotherapy treatment which dramatically reduce lymphoid cell numbers, making it virtually impossible to examine pre-cDC1s and CD8α+ cDC1s through cell isolation or adoptive transfer in this context. 

We have emphasized these specific points within our manuscript (Lines 254-257 and Lines 550-555).

2. We additionally have emphasized in our discussion that this work precipitated from previously published findings that found a strong association between the frequency of host pre-cDC1s and a reduction in death from lethal GvHD in a major-mismatched BMT model (Lines 642-645).

3. We have edited our manuscript to indicate that our functional assays are better described as ex vivo, as opposed to in vitro. This serves to clarify that all dendritic cells and T-cells used in functional assays were primary cells derived directly from living mouse tissue. Primary cells are recognized for closely mimicking in vivo cells by retaining the morphological and functional characteristics of their tissue of origin. The experimental use of primary cells therefore generates more relevant data representing living systems. 

4. We have also edited our manuscript to appropriately frame our findings within the borders of ex vivo experimentation and emphasize that our interpretations are not extrapolated to make conclusions about what occurs in vivo.

5. Lastly, we would like to acknowledge that working to develop rigorous methods to study these cells in vivo would provide compelling and important information. However, as the original submission of this manuscript was in August 2021, the lead author (MSM) who performed these experiments as part of her PhD work has since departed from the University of Arizona to complete her post-doctoral training at another institution. Additionally, the second and third authors who assisted in these studies have also moved to other positions. This unfortunately limits our ability to collect further data. 

Reviewer #2: Indicated that she/he did not find any issues that needed to be addressed and further complimented that our study provides important new findings in the field and that the manuscript can be accepted and published at its current state. We thank the reviewer for their time and positive remarks.

We thank you and the reviewers for taking the time to review our manuscript. We sincerely hope that our revisions have adequately addressed the minor comments of Reviewer #1. We look forward to hearing back from you soon and hope that our submission can be accepted without requiring additional re-review. As it has been nine months since our initial submission, we are excited to move forward with this publication. 

We thank you again for your time and careful consideration of our revised manuscript, and we look forward to your final decision.

Sincerely,

Megan Stanley Molina (meganm4@email.arizona.edu)

Emely A Hoffman (hoffmane@email.arizona.edu)

Jessica Stokes (stokes@email.arizona.edu)

Nicole Kummet (nkummet@email.arizona.edu) 

Richard J Simpson (rjsimpson@email.arizona.edu) 

Emmanuel Katsanis (ekatsanis@peds.arizona.edu)

---

## [Decision Letter · Decision Letter 1]

3 Aug 2022

Murine Precursors to Type 1 Conventional Dendritic Cells Induce Tumor Cytotoxicity and Exhibit Activated PD-1/PD-L1 Pathway

PONE-D-21-26682R1

Dear Dr. Molina,

We’re pleased to inform you that your manuscript has been judged scientifically suitable for publication and will be formally accepted for publication once it meets all outstanding technical requirements.

Kind regards,

Senthilnathan Palaniyandi, Ph.D

Academic Editor

PLOS ONE

Additional Editor Comments (optional):

Reviewers' comments:

Reviewer's Responses to Questions

**Comments to the Author**

1. If the authors have adequately addressed your comments raised in a previous round of review and you feel that this manuscript is now acceptable for publication, you may indicate that here to bypass the “Comments to the Author” section, enter your conflict of interest statement in the “Confidential to Editor” section, and submit your "Accept" recommendation.

Reviewer #1: All comments have been addressed

Reviewer #2: All comments have been addressed

2. Is the manuscript technically sound, and do the data support the conclusions?

Reviewer #1: Partly

Reviewer #2: Yes

3. Has the statistical analysis been performed appropriately and rigorously? 

Reviewer #1: Yes

Reviewer #2: Yes

4. Have the authors made all data underlying the findings in their manuscript fully available?

Reviewer #1: Yes

Reviewer #2: Yes

5. Is the manuscript presented in an intelligible fashion and written in standard English?

Reviewer #1: Yes

Reviewer #2: Yes

6. Review Comments to the Author

Reviewer #1: all concerns addressed

no further concerns

all concerns addressed

no further concerns

all concerns addressed

no further concerns

Reviewer #2: All previous comments have been addressed by the Authors. I thank the Authors for their work on the manuscript. I wish the best for all of the Authors' future endeavors.

7. PLOS authors have the option to publish the peer review history of their article (what does this mean?). If published, this will include your full peer review and any attached files.

Reviewer #1: **Yes: **nelson chao

Reviewer #2: No

---

## [Editor Report · Acceptance letter]

9 Aug 2022

PONE-D-21-26682R1 

Murine Precursors to Type 1 Conventional Dendritic Cells Induce Tumor Cytotoxicity and Exhibit Activated PD-1/PD-L1 Pathway 

Dear Dr. Molina:

I'm pleased to inform you that your manuscript has been deemed suitable for publication in PLOS ONE. Congratulations! Your manuscript is now with our production department. 

Kind regards, 

on behalf of

Dr. Senthilnathan Palaniyandi 

Academic Editor

PLOS ONE